# The Contribution of Tumor Derived Exosomes to Cancer Cachexia

**DOI:** 10.3390/cells12020292

**Published:** 2023-01-12

**Authors:** Christopher R. Pitzer, Hector G. Paez, Stephen E. Alway

**Affiliations:** 1Department of Physiology, College of Medicine, University of Tennessee Health Science Center, Memphis, TN 38163, USA; 2Integrated Biomedical Sciences Graduate Program, College of Graduate Health Sciences, University of Tennessee Health Science Center, Memphis, TN 38163, USA; 3Laboratory of Muscle Biology and Sarcopenia, Department of Physical Therapy, College of Health Professions, University of Tennessee Health Science Center, Memphis, TN 38163, USA; 4Center for Muscle, Metabolism and Neuropathology, Division of Regenerative and Rehabilitation Sciences, College of Health Professions, University of Tennessee Health Science Center, Memphis, TN 38163, USA; 5The Tennessee Institute of Regenerative Medicine, Memphis, TN 38163, USA

**Keywords:** cachexia, extracellular vesicles, exosomes, muscle, adipose, adipocytes, metabolism, inflammation

## Abstract

Cancer cachexia is defined as unintentional weight loss secondary to neoplasia and is associated with poor prognosis and outcomes. Cancer cachexia associated weight loss affects both lean tissue (i.e., skeletal muscle) and adipose tissue. Exosomes are extracellular vesicles that originate from multivesicular bodies that contain intentionally loaded biomolecular cargo. Exosome cargo includes proteins, lipids, mitochondrial components, and nucleic acids. The cargo carried in exosomes is thought to alter cell signaling when it enters into recipient cells. Virtually every cell type secretes exosomes and exosomes are known to be present in nearly every biofluid. Exosomes alter muscle and adipose tissue metabolism and biological processes, including macrophage polarization and apoptosis which contribute to the development of the cachexia phenotype. This has led to an interest in the role of tumor cell derived exosomes and their potential role as biomarkers of cancer cell development as well as their contribution to cachexia and disease progression. In this review, we highlight published findings that have studied the effects of tumor derived exosomes (and extracellular vesicles) and their cargo on the progression of cancer cachexia. We will focus on the direct effects of tumor derived exosomes and their cellular cross talk on skeletal muscle and adipose tissue, the primary sites of weight loss due to cancer cachexia.

## 1. Introduction

Cancer cachexia (CC) is broadly defined as the unintentional loss of body weight, primarily muscle mass, which may be accompanied by a loss of fat mass, in response to the progression of cancer. Cachexia is often preceded by anorexia; however, inflammation and metabolic hyper-catabolism are also prominent features of CC. This has led to strategies that use nutritional support as a key therapeutic approach for CC treatment, but these strategies are generally considered inadequate [1]. Skeletal muscle provides structural support and locomotive function to the body, which is an important part of maintaining a high quality of life. However, skeletal muscle also serves as the primary site of postprandial glucose uptake [2] and accounts for a large portion of the body’s caloric expenditure. Maintenance of muscle mass is a tightly regulated balance between muscle protein synthesis (MPS) and muscle protein breakdown (MPB). Both processes are needed for normal healthy muscle function [3], however in cancer cachexia, MPS is inhibited and MPB is stimulated resulting in muscle atrophy [4]. Adipocytes regulate whole body energy status by storing neutral lipids in the fed state, supplying those lipids to other tissues in the form of free fatty acids by lipolysis in times of unmet caloric need, and by regulating body temperature through thermogenesis in specialized fat depots [5]. The mechanisms underlying CC are incompletely understood. Tumor-stimulated inflammatory responses and catabolic factor release directly stimulate muscle loss; however, the metabolic purpose of muscle wasting in CC has not been determined. Nevertheless, skeletal muscle acts as a reservoir for the body’s amino acids [6]. It is therefore reasonable to speculate that muscle wasting occurs in CC to provide gluconeogenic precursors to accommodate the body’s energy needs.

Exosomes are extracellular vesicles with diameters ranging from 30–150 nm [7]. They contain a vast array of cargo molecules within a lipid bilayer that promotes survival in extracellular environments. Exosome cargo can include nucleic acids such as coding and noncoding RNAs and DNA, proteins, and lipids. The cargo may be expressed on exosome surfaces, contained within the exosomes, or anchored to intact or fragmented organelles including mitochondria [8,9,10,11]. Exosomal cargo biomolecules are intentionally packaged and altered in response to a vast number of stimuli, potentially allowing exosomes to be used to interpret the status of the secreting cell. Exosomes may enter the circulation or be transferred to nearby cells through the interstitial space and the biomolecular cargo may be taken up and alter signaling of other cells [8]. Neoplastic cells have altered inflammatory and hypoxia-related signaling and secrete exosomes at rates higher than noncancerous cells [12,13,14]. This has led to an interest in investigating the effects of tumor derived exosomes.

There are three main categories of exosomal cargo that will be discussed in this review. These include proteins, microRNAs (miRs), and circular RNAs (circRNAs). Like cells, exosomes exhibit proteins on the outer surface of their membrane bilayers, although proteins may also be contained within exosomes. One potential purpose of exosome secretion is removal of accumulating proteins in the secreting cell [15]. miRs are contained within secreted exosomes [16] and regulate gene expression through interactions with the 3′ untranslated region of mRNAs [17]. circRNAs serve multiple biological functions including serving as templates for protein synthesis; however, circRNAs also regulate the activity of miRs by sequestering them from mRNAs [18]. There are other components of exosomes and other types of cargo that may be contained within them, but the literature of tumor derived exosomes and their actions highlights these three components.

In this review, we aim to summarize the available literature concerning the contribution of tumor derived exosomes to the progression of CC. We will describe exosome secretion and potential stimuli prompting increased exosome release, the utility of exosomes as potential biomarkers, and the effects of tumor derived exosomes on skeletal muscle and adipose tissue. While both skeletal muscle and adipose tissue are also sources of circulating exosomes and other factors, for this review we will focus on the effects of tumor derived exosomes on these tissues.

## 2. Tumor Cell Derived Exosome Biogenesis, Cargo, and Release

### 2.1. Exosome Biogenesis

Virtually every cell type secretes exosomes, and as a result, exosomes are found in nearly every biofluid including the blood. There are two major types of exosome biogenesis. The most prevalent is dependent on the endosomal sorting complex required for transport (ESCRT complex) and is identified as ESCRT complex dependent exosome biogenesis. ESCRT complex independent exosome biogenesis occurs through the actions of other cellular machinery such as tetraspanin proteins and ceramides [19]. ESCRT complex dependent exosome biogenesis begins with endocytosis resulting in the formation of an early endosome. The endosome matures into a multivesicular body when further inward migration of the endosome membrane produces intraluminal vesicles. During this stage, the endosome is referred to as a multivesicular body (MVB). Intraluminal vesicles are intentionally loaded with cargo by the ESCRT complex, and the MVB either fuses with a lysosome to degrade the intraluminalvesicles or fuses with the plasma membrane to release intraluminal vesicles as exosomes. Once in the interstitial space, exosomes may be taken up by neighboring cells, interact with the extracellular matrix or be carried into the circulation [20]. Exosome biogenesis is an intricate process that is responsive to a multitude of biological inputs and a complete description of these processes is outside of the scope of this review. For a complete description of exosome biogenesis, see Gurunathan et al. [21].

### 2.2. Tumor Cell Derived Exosomes

The systemic nature of exosome release coupled with their broad range of effects on multiple organ systems and their known aberrant release by tumor cells makes exosomes a prime candidate to be investigated as mediators of CC. This is supported by the notion that tumor cell-conditioned media, which likely contains exosomes, causes dysfunction of muscle cells in vitro [22,23].

To our knowledge, there is no published evidence that shows that the fundamental biological processes of exosome biogenesis are altered in tumor cells compared to noncancerous cells. There are studies that point to various cancer types expressing increased ESCRT complex proteins [24], but it cannot be assumed that the results of these studies are indicative of all tumors, and overexpression of these proteins does not necessarily represent a change in the biological actions they perform. However, alterations in tumor cell biology that are common to many cancers may stimulate the process of exosome biogenesis; these conditions would similarly promote exosome release from noncancerous cells.

#### 2.2.1. Tumor Cell Derived Exosomes and Metabolic Remodeling in Response to Hypoxia

Hypoxia is defined as a decrease of oxygen availability in tissue and is a feature of many solid tumors due to defective perfusion as tumor cell proliferation outpaces angiogenesis. Markers of hypoxia may be indicative of tumor invasiveness [25]. Hypoxia induced factor-1 (HIF1) is a protein with expression that is tightly regulated by the abundance of oxygen in tissues. Cancer cells express HIF1 abundantly when perfusion needs are unmet and HIF1 regulates tumor cell metabolism in this condition [10]. Because oxidative phosphorylation by mitochondria requires oxygen as the final electron acceptor, it is not surprising that hypoxia causes upregulation of glycolytic metabolism in tumor cells. Consistent with this idea, Zheng et al. [26] report that exosomes derived from hypoxic pancreatic cancer cells stimulate glycolysis in normoxic pancreatic cells; a possible mechanism of chemotherapy resistance being conferred by HIF1 stimulated exosome release. HIF1 expression also stimulates macrophage glycolytic metabolism, which is a key regulator of proinflammatory M1 macrophage polarization [27]. Somewhat paradoxically, tumor derived exosomes have been shown to shift macrophages towards the anti-inflammatory M2 phenotype [28,29]. However, it must be considered that local immunosuppression in the tumor microenvironment promotes tumor cell survival [30].

Exosome release in response to hypoxia is a prosurvival physiological response; noncancerous cells show increased secretion of exosomes in hypoxia and the secreted exosomes likely serve to support cell survival in hypoxic tissues [31]. Umezu et al. [32] reported that hypoxic tumor cells released exosomes and the exosomal cargo in this context served to increase angiogenesis and reduce hypoxia. Because hypoxia stimulates exosome release in both cancerous and noncancerous cells, hypoxia stimulated exosome release should be interpreted to be a typical cellular survival mechanism rather than an aberrantly activated contributor to tumor pathogenicity. This fact, however, does not exclude tumor derived exosome as a potential therapeutic target for the treatment of CC. Figure 1A illustrates the stimulation of exosome release by hypoxia and the actions of these exosomes [33].

HIF1 expression and tumor derived exosomes released in response to hypoxia act in concert promote both global inflammation, as is typical of CC and tumor microenvironment alterations that promote tumor cell survival. These local and systemic responses to hypoxia alter metabolism throughout the body and represent a transition from tumor resistance to tumor tolerance; this concept is reviewed in detail by Maccio et al. [34].

#### 2.2.2. Cytokines and STAT3 Impact Exosome Release

Increases in circulating cytokines are a consistent observation of studies investigating CC. Interleukin-6 (IL-6) and tumor necrosis factor alpha (TNF-α) are commonly linked to the cachectic phenotype and directly suppressing muscle MPS [35,36,37]. Inflammatory cytokines such as IL-1, IL-6, and TNF-α cause the activating phosphorylation of Signal transducer and activator of transcription 3 (STAT3) [35,38]. STAT3 activation in skeletal muscle has been shown to be a central tenet of muscle wasting associated with CC [39]. While persistent STAT3 activation in muscle cells results in wasting, STAT3 hyperactivation is a feature of many tumor cell types [40]. Fan et al. [12] report that genetic manipulation of STAT3 in C26 tumor cells alters exosome release, pointing to a role of STAT3 activation in the increased rate of exosome secretion by tumor cells.

Exosome release is a cytoprotective mechanism during the onset of inflammation. Research outside the CC field indicates a potentially beneficial role of STAT3 mediated exosome release controlling proteostasis in Huntington’s disease [41]. Taken together, these facts imply that while there is no alteration to exosome biogenesis or release machinery, tumor cells are exposed to more intense stimuli that promote exosome release than noncancerous cells. Figure 1B illustrates the processes through which inflammatory mediators induce exosome release from tumors and the effects of those exosomes on adipose and skeletal muscle tissue.

#### 2.2.3. Exosomal Cargo

Exosomes are thought to act as an extra-endocrine means of intracellular communication. Exosome cargo can include proteins, lipids, nucleic acids, and mitochondria [9,11]. Exosomal cargo becomes altered to reflect the homeostatic status of the cells secreting the exosomes and may serve as a means for cells to remove toxic accumulations in their own cytoplasm [42]. Exosomal cargo is thought to alter the signaling of recipient cells through direct entry into cells [43], or through interactions with the extracellular matrix [44]. This alteration of signaling plays a role in the regulation of metabolism [45], extracellular matrix remodeling [44], or angiogenesis [46], and may serve as a modulator of innate immunity [47]. These functions are important for the progression of cancer and the development of cachexia. Furthermore, tumor cells secrete exosomes abundantly, and tumor derived exosomes likely mediate the metastatic cascade and alterations in tumor microenvironment [48]. The observation that exosomes are involved in cancer progression supports the notion that exosomes may contribute to tumor growth by supporting tumor cell survival. 

### 2.3. Exosome Release from Tumor Cells as a Potential Therapeutic Target

Literature illustrating that tumor cells secrete exosomes in abundance raises questions concerning the biological activity of tumor derived exosomes, and by extension the contribution of tumor derived exosomes to CC. Exosomes that are released from noncancerous cells generally function to promote cell survival. The molecular triggers for exosomal release from tumor cells, appear to have similar mechanisms to those regulating release of exosomes of noncancerous cells. Therefore, it stands to reason that inhibition of exosome release from tumor cells may reduce tumor function or survival and improve CC associated outcomes. To this end, there is an emerging body of literature that utilizes tumor cell exosome release as a potential therapeutic target. Liu et al. [49] showed that omeprazole inhibited an increase in circulating exosome content seen in Lewis lung carcinoma (LLC) inoculated mice. Treatment using omeprazole also significantly improved muscle fiber cross sectional area and survival of LLC inoculated mice [49]. GW4869, a pharmaceutical agent that inhibits exosome formation, has been used to show that tumor derived exosomes play a role in B-cell lymphoma 2 (BCL-2) mediated apoptosis initiation in skeletal muscle cells [50] and adipose tissue browning [51]. Zhou et al. [52] report that amiloride, a potassium sparing diuretic drug, inhibits exosome release from both LLC and C26 tumor bearing mice, which prevented declines in tibia-length normalized muscle wet weight (a measure of muscle mass) and improved grip strength (a measure of muscle function). Together, these studies suggest that tumor derived exosomes play a role in wasting of both muscle and adipose tissue, and blocking their release may reduce the severity of CC.

## 3. Exosomes as a Biomarker of Cachectic Cancers

It has been established that tumor cells release exosomes at a greater rate than healthy cells [53]. Given the context dependent nature of exosome cargo loading, it is not surprising that exosome cargo from tumor cells differs from that of nonmalignant cells. Several studies have aimed to take advantage of this fact and identify circulating exosomal cargo that may be used as biomarkers. Because different tumor types have different causes and paraneoplastic outputs, there is not likely to be a single biomarker for presence of tumor or CC progression. However, cancers of similar types have been found to secrete specific exosomal cargo that correlate to the presence of tumors, tissue wasting, and other variables.

### Exosome Micro RNAs (miRs) Are Linked to Tumor Presence and Muscle Wasting

Cancer cells secrete exosomes abundantly and stimulate the release of exosomes from other tissues. In general, the published literature points to exosomal cargo and circulating nucleic acids as promising biomarkers for tumor presence and development of CC in multiple types of cancer. For example, Yamada et al. [54] found that presence of miR-21, miR-29a, and miR-125b in human serum were indicative of the presence of colorectal tumors, and that these miRs were elevated even at early stages of neoplasia. Follow-up work from this group found that the levels of circulating miR-21 was inversely correlated to psoas muscle mass index, implicating miR-21 as a potential biomarker for colorectal cancer cachexia [55]. While miRs were likely part of the exosomal cargo, the limitation of this observation is that neither of these studies demonstrated unequivocally that the circulating miRs linked with cancer cachexia were contained within exosomes. This is because these studies used non-fractionated serum samples in their analysis, and as such, they did not deplete the serum of exosomes to remove exosome bound miRs from the analysis. Nevertheless, it is reasonable to speculate that tumor cells either directly secrete these miRs or stimulate their release from other tissues. In either case, exosomal release is a major pathway involved in extracellular export of miRs [56]. The published literature supports the notion that a measurable portion of circulating miRs are contained within exosomes and that cancers of a similar origin will cause similar miR secretory patterns. It is generally believed that exosome packaging serves to improve stability and survivability in the extracellular space and therefore prolongs the potential impact of miRs on cellular function. Destabilizing exosomal packaging or regulating exosomal miR cargo represents a promising opportunity in the field of personalized medicine to offset CC.

## 4. Contribution of Exosomes to Skeletal Muscle Inflammation in Cancer Cachexia

### 4.1. Exosomal Cytokine Cargo

Despite a clear component of anorexia in the development of cachexia, increased inflammatory cytokine circulation likely precedes loss of appetite and plays a causal role in CC associated weight loss [57]. Several cytokines have been shown to exhibit either pro-cachectic or anti-cachectic actions [58]. It is generally assumed that cytokines are released into circulation directly, however Hu et al. [59] report that depletion of extracellular vesicles from LLC cell conditioned media decreased STAT3 activation and expression of atrogin-1 (an E3 ubiquitin ligase frequently expressed during muscle atrophy). While this study does not directly illustrate the presence of IL-6 in exosomes, it does show that extracellular vesicle depletion from conditioned media decreases the inflammatory stimulus. Dalla et al. [60] report that cytokines are packaged within breast cancer derived exosomes and that the concentration found in secreted exosomes correlates to concentrations of those cytokines found in tumor cell conditioned media.

Evidence that tumor derived exosomes may package inflammatory cytokines, indicates a potential role of exosomes in the activation of STAT3 signaling. STAT3 is a transcription factor that is responsive to multiple inputs, notably Interleukin-6 (IL-6) and TNF-α [61]. Multiple receptors serve to activate STAT3 including interleukin receptors, GP130, and TGFβ receptors. Activated STAT3 is phosphorylated and forms a homodimer that enters the nucleus and activates transcription of genes favoring MPB and satellite cell activation [62]. STAT3 activation stimulates MPB through activation of the ubiquitin proteasome system (UPS) and by upregulating E3 ubiquitin ligases MuRF1 and atrogin-1 in skeletal muscle [63] as well as through activation of nuclear factor kappa-light-chain-enhancer of activated B cells (NF-κB) [61]. Supporting the notion that tumor derived exosomes may activate STAT3 in muscle cells, Ham et al. [64] report that exosomes derived from breast cancer cells activate GP130/STAT3 signaling and drive immunosuppressive M2 macrophage polarization.

### 4.2. Exosomal Heat Shock Protein Cargo

Tumor derived exosomes are an established effector of the tumor microenvironment and their effects have been shown to play a role in metastasis as well as tumor cell survival [65]. Heat shock proteins (HSPs) are molecular chaperone proteins that regulate protein stability, folding, and protein complex assembly/disassembly [66]. Tumor derived exosome membranes have been reported to be enriched in various HSPs since the mid-2000s. This is not surprising because tumor cells generally overexpress heat shock proteins [67]. HSP expression in tumor derived exosomes has been linked to increased cancer cell motility [68], and suppression of antitumor immune reaction [69]. Interestingly, HSP mediated immune suppression in tumor derived exosomes has been shown to operate in a STAT3 dependent manner [70]. STAT3 activation in this context is potentially mediated through Toll Like Receptor 4 (TLR4) [71]. Niu et al. [72] show that pharmacological inhibition of HSP90 in mice prevented tumor-associated muscle wasting. Furthermore, HSP90 increased the duration of activation of STAT3 [72]. Persistent STAT3 activation in the presence of HSP90 is a possible intracellular consequence of HSP delivery to the muscle cell by tumor derived exosomes. Consistent with this hypothesis are data from Liu et al. [49], who show that omeprazole treatment prevents the increases in circulating HSP70 and HSP90 in response to tumor inoculation. Omeprazole treatment also preserved muscle mass and function as well as improved survival of the muscle cells. Together, the published literature illustrates that tumor cells overexpress HSPs as a pro-survival response. These HSPs are packaged into exosomes where they influence the tumor microenvironment, inhibit immune reactions to tumor cells, and initiate pro-cachectic STAT3 activation in skeletal muscle.

We recognize that other inflammation responses occur in the skeletal muscle during CC. However, this review is focused on the actions of exosomes and extracellular vesicles in mediating CC; therefore, a broad discussion of other inflammatory responses that have not been shown to be associated with exosomes or mediation of responses via an exosomal mechanism are outside the scope of this review. For further review on the widespread effects of inflammation on skeletal muscle during CC see Webster et al. [73].

## 5. Noninflammatory Effects of Cancer Cachexia Associated Exosomes on Skeletal Muscle

### 5.1. Noninflammatory Actions of miRs

CC is an inherently inflammatory process; however, the pleiotropic nature of many of the mediators of systemic inflammation stimulate noninflammatory processes that are also detrimental to the muscle. Skeletal muscle is heavily enriched in several miRs, these are collectively referred to as myomiRs [74]. MyomiRs have been shown to play a role in mediating atrophy through multiple pathways [74]. miRs contained within tumor-derived exosomes have been shown to mediate aspects of CC in various types of cancer through multiple mechanisms. Qiu et al. [75] implicated miR-181a-3p in oral squamous cell carcinoma exosomes as an activator of endoplasmic reticulum stress, muscle atrophy, and apoptosis in vitro and in vivo. Miao et al. [50] reported that two distinct miRs (miR-195a-5p and miR-125b-1-3p) were upregulated in exosomes of C26 colon cancer tumors. Furthermore, C26 tumor derived exosomes initiated muscle atrophy in vivo and in vitro and miR mimic treatment of either of the identified miRs was sufficient to induce myotube atrophy and apoptosis [50]. These studies illustrate a clear role of tumor derived exosomes in initiating skeletal muscle cell apoptosis during CC.

### 5.2. Cancer Cachexia Associated miRs Alter Muscle Metabolism

Outside of apoptosis, tumor derived exosome miRs have been shown to alter various facets of skeletal muscle metabolism. Wu et al. [76] have shown that breast cancer derived exosomes contain greater levels of miR-155 and enhanced anaerobic glycolysis in C2C12 myotubes after inhibiting Peroxisome proliferator-activated receptor gamma (PPARγ). This study also found increased lipolysis, which is also prevalent in CC [76]. It should be noted that PPARγ is known to regulate mitochondrial metabolism as well as activate PGC1α, which has been identified as a “master regulator of mitochondrial biogenesis” [77,78]. This would indicate that these exosomes primarily stimulate glycolytic metabolism. Conversely, exosomes derived from pancreatic cancer cells express miRs that induce insulin resistance and inhibit glucose metabolism [79]. These studies highlight the heterogenous nature of cancers and the exosomes that they may secrete. Breast and pancreatic cancers are both known to cause CC, however the studies discussed above show that their exosomes exert opposing effects on glucose and mitochondrial metabolism in skeletal muscle. This indicates that there is a need for establishing a degree of personalized medicine when considering exosome release as a potential target of novel treatment strategies for CC.

### 5.3. Exosomal Proteins in Cancer Cachexia

Proteins are found both in exosomal cargo and on exosome membranes [80]. Functional proteins carried in exosomes are delivered to recipient cells where they exert their physiological actions [80]. Growth and differentiation factor 15 (GDF15) is a protein of recent interest in the literature because it has been shown that its’ expression is inversely correlated with muscle mass and endurance [81]. Recently, Zhang et al. [82] demonstrated that GDF15 induces myotube atrophy and induces BCL-3/caspase-3 mediated apoptotic pathways in muscle cells. It was interesting to note that this study reported that extracellular GDF15 (by treating culture media with recombinant protein) or intracellular GDF15 (contained within exosomes) stimulated atrophy to a similar degree [82]. Other data show that prostate cancer may stimulate the release of GDF15 from osteocytes to alter the tumor microenvironment and promote metastasis [83]. To our knowledge, there are no published studies that have identified the causal mechanism for increased exosomal GDF15 release from tumor cells. However, HIF1 has been implicated in stimulating GDF15 release in hypoxic osteoblasts [84]. It is therefore reasonable to speculate that HIF1 stimulates both GDF15 expression and exosome release in tumor cells, resulting in increases in circulating GDF15 that contributes to CC.

## 6. Effects of Tumor Derived Exosomes on Adipose Tissue

### 6.1. White and Brown Adipose Tissue

There are two major types of adipose tissue in the body, white adipose tissue (WAT), and brown adipose tissue (BAT). WAT serves as the primary lipid storage depot (contributing to the white coloration). In times of caloric need, WAT will undergo lipolysis and send energy substrates to other cells of the body in the form of free fatty acids. WAT exists in two primary fat depots in the body; visceral adipose tissue (VAT), which surrounds the organs in the abdominal cavity, and subcutaneous adipose tissue (SAT) which resides under the skin [85]. BAT is much more vascularized (contributing to the brown coloration) than WAT and serves to regulate whole body thermogenesis and energy expenditure. This function is linked to a greatly increased expression of uncoupling protein 1 (UCP1) which decreases metabolic efficiency and converts the chemical energy to heat [86]. BAT generally exists in strategic locations to protect against hypothermia [87]. Apart from these classically described functions of adipose tissue, more recent data suggest that adipose tissue is an important secretory organ and it plays an important role in endocrine regulation of metabolism [88].

### 6.2. Exosomal Regulation of Adipogenesis

Adipose tissue contains a population of cells known as preadipocytes which may differentiate to form mature adipocytes when necessary [89]. This process is referred to as adipogenesis. The literature points to three primary effects of tumor derived exosomes on adipocytes: increased “browning” of WAT, aberrant lipolysis, and inhibition of adipogenesis [90]. Inhibition of adipogenesis with simultaneous upregulation of lipolysis will cause atrophy of adipose tissue that is unlikely to be reversed unless the cachectic stimulus is resolved. Tumor derived exosomes have been shown to enhance lipolysis as well as inhibit adipogenesis [59,91]. It is important to note that adipose tissue wasting is not present in all cases of CC observed clinically. Conversely, in cases of CC with simultaneous adipocyte and myocyte wasting, adipocyte atrophy tends to occur prior to myocyte atrophy and reversing adipocyte atrophy may be protective of myocyte atrophy [92].

### 6.3. Tumor Derived Exosomes Contribute to Adipose Tissue Browning

Adipose tissue browning is characterized by an increase in vascularization, increased expression of mitochondrial UCP1, upregulation of mitochondrial content and the presence of multilocular lipid droplets [93]. The adipose tissue phenotype is adaptive and responsive to many stimuli, including cold exposure [94,95], exercise [96], and IL-6 [97]. As described above, IL-6 and other proinflammatory cytokines may be present in tumor derived exosomes. Furthermore, some portion of the IL-6 that is secreted from adipocytes in CC likely comes from circulating exosomes [59]. Elevated browning of adipocytes in response to CC-induced IL-6 exposure is to be expected, because it has been shown that increased circulating IL-6 levels in response to exercise is a primary mechanism inducing browning of adipocytes [98,99]. Furthermore, Knudsen et al. show that this IL-6 response directly mediates adipose tissue browning [100]. Chronic IL-6 elevation, independent of IL-6 packaging into exosomes, likely contributes to pathological WAT browning in the CC context.

#### 6.3.1. Exosomal Noncoding RNAs

The gene expression signature of BAT is markedly different from that of WAT, leading to the differences in tissue phenotype [101]. Noncoding RNAs are a known component of the exosome cargo that may alter gene expression [102]. Therefore, it is reasonable to expect that in addition to IL-6, tumor derived exosomes alter their gene expression to promote the adipose tissue browning observed in CC. Hu et al. [51] addressed this hypothesis and reported that inhibition of tumor derived exosome generation using GW4869, suppressed adipose tissue browning in vitro and in vivo. While this study illustrated that tumor derived exosomes contributed to adipose tissue browning [51], it is limited to the LLC type tumor cell and did not investigate specific exosome cargo to determine the mechanism of action. 

Given the heterogeneity of cancer pathologies, it is not surprising that different types of tumors have a range of tumor derived exosomes that contain different cargo that result in adipocyte browning. Zhang et al. [103] implicated gastric cancer derived exosomal circRNA ciRS-133 in adipose tissue browning by directing the differentiation of preadipocytes to a brown phenotype. Colorectal cancer derived exosomal miR-146b-5p acting through Homeobox C10 (HOXC10) has also been shown to elicit adipose tissue browning [104]. Exosomes in non-small cell lung cancers were shown to initiate white adipose tissue browning through the actions of miR-425-3p [105]. miR-425-3p has also been reported to inhibit preadipocyte proliferation and differentiation [105]. Collectively, the published literature shows that while there is likely no single noncoding RNA exosome cargo signature that is common to all cancers that results in adipose tissue browning, there is growing evidence to indicate that tumor derived exosomes initiate adipocyte browning via the noncoding RNA exosome cargo.

While there is no single consistent molecule in CC that causes adipose tissue browning, the literature indicates a more common metabolic effect of tumor derived exosomes on adipose tissue than in skeletal muscle. Adipose tissue browning indicates the potential for increased mitochondrial fatty acid oxidation and mitochondrial biogenesis in response to tumor derived exosomes. This may represent a potential therapeutic opportunity for CC, as BAT has a much higher metabolic rate than WAT and hypercatabolic metabolism is the hallmark phenotype of CC.

#### 6.3.2. Endocrine Regulation by Exosomes

Adipose tissue has an established role as an endocrine organ and is known to be responsive to multiple endocrine stimuli [106,107]. Parathyroid hormone-related peptide (PTHrP) is a peptide hormone known to be secreted by some cancers [108], likely as an angiogenic factor [109]. Hu et al. [110] have recently shown that PTHrP may be contained within LLC tumor derived exosomes and may also contribute to adipocyte browning in vivo. Interestingly, LLC tumor derived exosomes expressed PTHrP, but C26 tumor derived exosomes did not [110]. While proteins are a frequent component of exosomes in both cancerous and noncancerous cells, there is little evidence in the literature to support the idea that hormones are otherwise frequently contained in exosomes. Nevertheless, the presence of PTHrP in LLC secreted exosomes likely indicates an abundance of PTHrP in the cytosol of LLC cells. This raises questions about other potential effects of proteins associated with tumor derived exosomes on the adipose tissue phenotype. This question warrants further investigation.

### 6.4. Effects of Tumor Derived Exosomes on Lipolysis and Adipose Tissue Wasting

Lipolysis is the process in which adipocytes catabolize the neutral lipids contained within their fat droplets to free fatty acids for release into the blood where they may be taken up and metabolized by other cells in the body [111]. The process of lipolysis is sequentially mediated by a class of proteins referred to as lipases. There are 3 lipases in humans (ATGL, HSL, and MGL), and each lipase is responsive to different stimuli and inhibitors [112]. The rate of lipolysis is increased in BAT in response to metabolic activation such as cold exposure [113], and lipolysis is heavily upregulated in CC in which adipose tissue wasting is observed [114]. Given our current understanding of exosome mediated metabolic alterations, and the participation of exosomes in the browning of adipose tissue, it is reasonable to expect that tumor derived exosomes play a role in upregulating lipolysis in CC. In fact, the literature that attributes adipose tissue browning to tumor derived exosomes shows increased rates of lipolysis in response to exosome treatment.

### 6.5. Exosome Regulation of Adipocyte Metabolism

miRs are known to participate in the regulation of metabolism through altering gene expression of proteins that regulate metabolic enzymes, or by altering the expression of the metabolic enzymes themselves [115]. A hypercatabolic metabolism is a defining feature of CC, and tumor derived exosomal noncoding RNAs are prime candidates for investigation of this phenomenon. Several miRs that have already been discussed have been shown to stimulate lipolysis through multiple mechanisms: miR-155 in breast cancer, miR-146b-5p in colorectal cancer, and miR-425-3p in non-small cell lung cancers all induce adipose tissue wasting by upregulating lipolysis [104,105,116]. While these studies show that the addition of multiple miRs to tumor derived exosomes in multiple cancer types stimulate lipolysis, there is an appreciable gap in the published literature concerning the downregulation of miRs that may also occur. Stimulation of metabolic processes may come from inhibition of inhibitors (e.g., a miR blocking the translation of an mRNA whose product would inhibit the action of another molecule), or by the absence of an inhibitory stimuli and allowing translation of mRNAs that would otherwise be inhibited.

### 6.6. Cytokine-Associated Exosome Regulation of Adipocytes

It is clear from the literature that exosomes and their cargo act in a pleiotropic manner to induce the CC phenotype. This is especially true of exosomal proteins, which often act on both skeletal muscle and adipose tissue to cause wasting. One such example is IL-6, which has been implicated in the wasting of skeletal muscle tissue through its activation of STAT3 (see above). The same pathway is activated for adipose tissue, in which LLC derived exosomal IL-6 induces lipolysis [59]. The effects of chronic IL-6 elevation are similar in skeletal muscle and adipose tissue because they activate the same downstream pathways. Sagar et al. [117] show that adrenomedullin is upregulated in pancreatic cancer derived exosomes, and stimulates lipolysis in both humans and mice. Interestingly, adrenomedullin contributes to β-cell dysfunction in the pancreas resulting in diabetes, secondary to neoplasia [118]. This mimics the phenotype observed in Type 1 Diabetes, which causes hyperglycemia induced muscle wasting [119].

### 6.7. miR-Associated Exosome Regulation of Adipogenesis and Lipolysis

Inhibition of adipogenesis is coupled with stimulation of lipolysis in CC to result in wasting of adipose tissue. Tumor derived exosomes have been shown to inhibit adipogenesis through multiple miR-mediated mechanisms. Wan et al. [91] have shown that chronic myeloid leukemia cells (K562) and their secreted exosomes expressed high levels of miR-92a-3p, which inhibited the expression of CCAAT/enhancer binding protein alpha (C/EBPα) transcripts. This study also found that miR-92a-3p expression was upregulated in other tumor types including MCF-7, Raji, HT-29, A549, and U118. Exosomal miR-410-3p has been implicated in gastric cancers as an inhibitor of adipogenesis through inhibition of IRS-1 [120]. IRS-1 is a critical component of insulin signaling and is important for insulin stimulated glucose uptake in the muscle [121]. This represents a potential mechanism through which muscle atrophy (through defective glucose uptake) and adipose atrophy (through inhibited adipocyte differentiation) may occur simultaneously.

## 7. Conclusions

The published literature (summarized in Table 1) suggests that tumor derived exosomes stimulate loss of MPS, and an increase in MPB, while also leading to adipose tissue browning, as well as enhancing lipolysis and inhibit adipogenesis. This leads to wasting of muscle and adipose tissue and depletion of the body’s calorie storage and contributes to the regulation of CC (Figure 2).

The overwhelming majority of the published literature supports a role for exosomal regulated CC (Table 1). However, there is a smaller subset of literature that suggests that exosomes may not play all of the physiological roles that have been attributed to them. For example, Albanese et al. [122] report that only a minority of circulating miRs are contained within exosomes and that a bulk of them are actually complexed to proteins. Albanese and colleges [122] further argue that the miR content that has been identified in exosomes is potentially too small to efficiently regulate gene expression. Although the level of miRs needed to efficiently regulate gene expression is not known, these experiments raise some doubt regarding the necessity of miRs to be packaged into exosome membranes to enter the circulation. Nevertheless, these few studies do not refute the large number of studies that show that circulating nucleic acids can be predictive of tumor presence and correlate with CC outcomes [123].

A similar concern has been raised as to whether exosomes regulate inflammatory responses in cancer that lead to CC. Benjamin-Davalos et al. [124] argue that contaminating inflammatory soluble factors are prevalent in all frequently used exosome isolation procedures in research. They argue that this may impact the interpretation of downstream results and exaggerate the roles of exosomes in driving inflammatory processes [124]. Resolution of this question is a current limitation of the study of exosome related physiology. It is important to note that inflammatory components are also packaged within exosomes or expressed on the outside of exosome membranes [125], further complicating this problem.

While overall, there is compelling and mounting evidence that biomolecules that are also frequently contained within exosomes exert biological effects that contribute to CC, or are potential biomarkers of tumor presence or prognosis, this conclusion is not shared by all investigators in the field. Whether these molecules are contained within exosomes in every case is a current gap in the field of exosome research, and is an important question to resolve if exosome targeting is to become a viable strategy to counter CC.

## 8. Future Directions

There is an emerging scientific interest in using bioengineered exosomes to deliver specific biomolecules that would otherwise be unstable in circulation to cells throughout the body [126,127,128]. Exosomes are uniquely suited to this purpose because they are biologically equipped to secrete proteins that accumulate in the cytoplasm. Future research regarding exosomes impacting CC, should utilize bioengineering to attempt to deliver biomolecules that will oppose the actions of tumor derived exosome cargo, possibly by increasing the cytosolic expression of biomolecules that are to be packaged into exosomes. These engineered exosomes may also attempt to replace downregulated exosomal cargo that are found in decreased abundance in the circulation in response to tumors. Bioengineered exosomal cargo has the strong potential to address unmet needs in the body in response to tumor development to attenuate or prevent CC.

Future research into exosomes should focus on the development of isolation methods that increase exosome yield and purity. This will improve confidence in research regarding the biological effects that exosomes exert on tissues. Removal of cytokines, hormones and other biologically active substances from exosome preparations will allow for experimental delineations of the effects of exosomes themselves compared to the effects of other circulating factors in disease. Pure isolations of whole intact exosomes will also allow for increased clarity regarding the presence and localization of potential exosomal cargo molecules. Thus, improving the quality of exosome preparations will help to increase the confidence in the experimental findings and in doing so, better address the scientific research gaps, and to clarify the importance and roles of exosomes in directly mediating CC.

## Figures and Tables

**Figure 1 cells-12-00292-f001:**
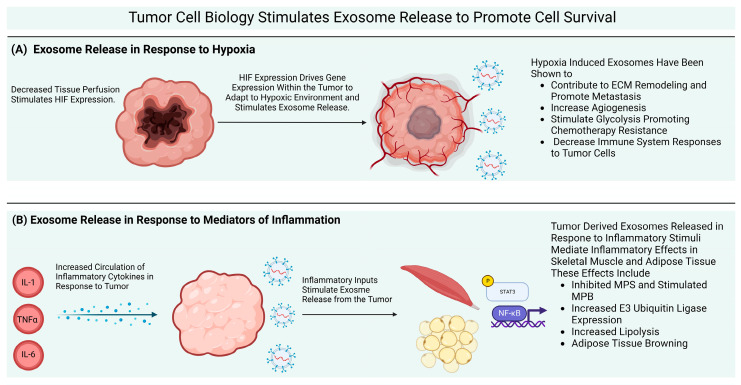
Exosome release in noncancerous cells is generally a pro-survival mechanism that is meant to relay signals of cellular stress throughout the body. Exosome release from tumor cells serves the same purpose, however tumor cells are exposed to a multitude of cellular stresses that stimulate exosome release. These exosomes interact with the tumor microenvironment and enter circulation where they exert systemic effects. (**A**) Tumor derived exosome release is stimulated by hypoxia to promote cell survival by altering the ECM components of the tumor microenvironment, stimulating angiogenesis, altering tumor and host cell metabolism, and altering local macrophage polarization. (**B**) Circulating inflammatory mediators also stimulate exosome release from tumor cells. These exosomes directly impact the MPS/MPB balance, increase E3 ubiquitin ligase expression, increase lipolysis and cause adipose tissue browning. Collectively, these effects illustrate a clear role of tumor derived exosomes in supporting cancer cell survival and contributing to the development of CC. Exosomes represent a novel potentially important therapeutic target for countering CC.

**Figure 2 cells-12-00292-f002:**
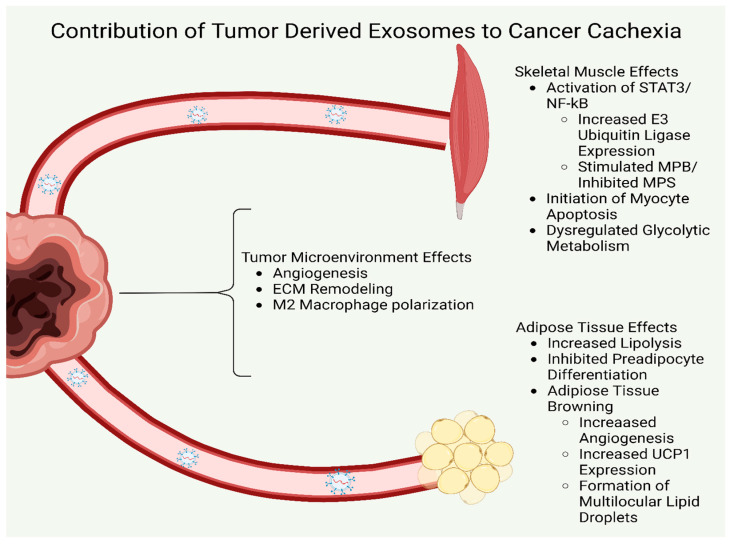
Exosome regulation of muscle and adipose wasting in cancer cachexia. Tumor cells secrete a range of exosomes which differ based on the tumor type. Nevertheless, common features of exosomes include a tumor regulated inflammatory environment which stimulates cytokine and E3 ligase activation in skeletal muscle along with an upregulation of signaling for apoptosis, reduced myocyte protein synthesis, and increased muscle protein degradation. Furthermore, tumor derived exosomes stimulate a dysregulated metabolic phenotype in skeletal muscle. Similarly, tumor derived exosomes decrease adipogenesis, increase browning of adipose cells and increase lipolysis, which contributes to overall loss of adipose stores. The resultant loss of skeletal muscle and adipose tissue is defined as cancer cachexia.

**Table 1 cells-12-00292-t001:** A summary of the studies that describe the effects, and important cargo molecules of tumor derived exosomes on the body.

Secreting Cell Type	Cancer Type	Key Exosome Cargo	Target Cell	Biological Actions of Exosomes	Citation
TCGA Database lung cancer cells	Lung	↓ miR101	Macrophages	↑ inflammation	[13]
BXPC-3 SW1990	Pancreatic	↑ CircZNF91	No Target	↓ miR-23b-5p ↑ glycolysis ↑ chemotherapy resistance	[26]
PANC-1 BXPC-3	Pancreatic	↑ miR-301a-3p	Macrophages	↑ M2 macrophage polarization ↑ tumor migration/invasion	[28]
HCT-8 LOVO	Colorectal	↑ miR-934	Macrophages	↑ M2 macrophage polarization ↑ metastasis	[29]
RPMI8226 KMS-11 U266	Multiple Myeloma	↑ miR-135b	ECM	↓ FIH1 ↑ angiogenesis	[32]
U373	Glioblastoma multiforme	↑ CRYAB	No Target	↑ tumor cell apoptosis resistance	[37]
LLC	Lung	↑ HSP70 ↑ HSP90	C2C12 myotubes/mouse muscle	↑ myotube/muscle catabolism	[49]
Human Colon Tumors C26	Colorectal	↑ miR-195a-5p ↑ miR125b-1-3p	C2C12 myotubes/mouse muscle	↑ myotube/muscle catabolism	[50]
LLC	Lung	No Specific Cargo	3T3-L1 Adipocytes	↑ lipolysis	[51]
CT26 LLC	Colon Lung	No Specific Cargo	C2C12 myotubes/mouse muscle	↑ myotube/muscle catabolism	[52]
Human Colorectal Tumors	Colorectal	↑ miR-21	Muscle	↑ psoas muscle wasting—prognostic indicator of survival	[55]
LLC	Lung	↑ IL-6	C2C12 myotubes 3T3-L1 adipocytes	↑ myotube atrophy ↑ lipolysis ↑ STAT3 phosphorylation	[59]
MCF-7 MDA-MB-231 T47D	Breast	↑ various proteins associated with tumor metastasis, formation, angiogenesis and immunotolerance	Multiple targets	Purposefully loaded exosome protein cargo varies by tumor type	[60]
EO771	Breast	↑ GP130	Macrophages	↑ M2 macrophage polarization ↑ tumor survival	[64]
A172 HT1080 MDA-MB-231 SUM159	Glioma Fibrosarcoma Breast Breast	↑ HSP90α	ECM	↑ MMP2 activation ↑ tumor motility	[68]
RENCA	Renal Cell Carcinoma	↑ HSP70	Myeloid Derived Suppressor Cells	↑ immune suppression	[70]
SCC7 SCC25 CAL27	Oral Squamous Cell Carcinoma	↑ miR-181a-3p	C2C12 myotubes	↑ muscle cell atrophy ↑ muscle cell apoptosis ↑ ER stress	[75]
Human Breast Tumors	Breast	↑ miR-155	Adipose Muscle	↑ PPARγ ↑ adipose browning	[76]
Mouse Pancreatic Cancer Cells	Pancreatic	↑ 9 miRs	C2C12 myo-tubes	↓ insulin signaling	[79]
C26	Colon	↑ GDF15	C2C12 cells	↑ muscle atrophy ↑ apoptotic signaling	[82]
Human Leukemia Cells	Chronic Myeloid Leukemia	↑ miR-92a-3p	Adipose derived mesenchymal stem cells	↓ adipogenesis ↓ CEBPα	[91]
Human Gastric Tumors	Gastric	↑ ciRS-133	Adipose	↓ miR-133 ↑ PRDM16 activation ↑ WAT browning	[103]
Human Colorectal Tumors	Colorectal	↑ miR-146b-5p	Primary white adipocytes	↓ HOXC10 ↑ WAT browning	[104]
A549 H1299 AGS	Lung Lung Gastric	↑ miR-425-3p	Human visceral preadipocytes and mature adipocytes	↓ preadipocyte proliferation and differentiation ↑ adipocyte lipolysis	[105]
LLC	Lung	↑ PTHrP	3T3-L1 adipocytes	↑ lipolysis ↑ WAT browning	[110]
Human Breast Tumors	Breast	↑ miR-155	C2C12 myotubes 3T3-L1 adipocytes	↑ tumor invasiveness ↑ WAT browning ↓ PPARγ	[116]
Human Gastric Tumors	Gastric	↑ miR-410-3p	Primary human white adipocytes	↓ IRS1	[120]

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
