# Peer review of "The Contribution of Tumor Derived Exosomes to Cancer Cachexia"

_cells, 2023, doi:10.3390/cells12020292_

Round 1
Reviewer 1 Report
Authors have described the effects of tumor derived exosomes (and extracellular 26 vesicles) and their cargo on the progression of cancer cachexia. Paper is fairly written and can be accepted if modifications are incorporated as follows:
· Please enlist the cargoes present in the exosomes. (Line – 55)
· Please elaborate the process of biogenesis. There are basically three different ways involved in biogenesis of exosomes. (Line – 81)
· Please incorporate the detailed table which contains list of exosomes, its origin, target molecule and target organ.
· What is the role of this exosomes on mitochondrial biogenesis and function? What is the impact of tumor derived exosomes on mitochondria of skeletal muscle and adipocytes?
· Is there any detail available regarding role of tumor or host derived exosomes on hypothalamus-pituitary axis?
Author Response
Response to Reviewers: The Contribution of Tumor Derived Exosomes to Cancer Cachexia
Reviewer #1
Thank you for your review and helpful comments for improving our manuscript.
Reviewer #1 comment. Authors have described the effects of tumor derived exosomes (and extracellular 26 vesicles) and their cargo on the progression of cancer cachexia. Paper is fairly written and can be accepted if modifications are incorporated as follows:
Authors response: We would like to thank the Reviewer for their constructive feedback and address the issues presented below.
Reviewer #1 comment. Please enlist the cargoes present in the exosomes. (Line – 55)
Authors response: A description of the various types of biomolecules contained within exosomes and on exosome surfaces has been added alongside relevant citations in this location.
Reviewer #1 comment. Please elaborate the process of biogenesis. There are basically three different ways involved in biogenesis of exosomes. (Line – 81)
Authors response: We have elaborated on ESCRT complex dependent exosome biogenesis, the most prevalent type of exosome biogenesis in mammalian cells. However, the basic properties of exosome biogenesis are not fundamentally changed in tumor cells and therefore we believe a complete discussion of exosome biogenesis is outside the scope of this review. Instead, we refer the reader to another review article detailing the multiple mechanisms of exosome biogenesis and the potential clinical applications of exosomes.
Reviewer #1 comment. Please incorporate the detailed table which contains list of exosomes, its origin, target molecule and target organ.
Authors response: This table has been constructed as suggested and it was added within the manuscript.
Reviewer #1 comment. What is the role of this exosomes on mitochondrial biogenesis and function? What is the impact of tumor derived exosomes on mitochondria of skeletal muscle and adipocytes?
Authors response: This is a very intriguing question and one that we wish we could answer fully. There is no consistent answer to this question from the published literature, as different cancer types and the exosomes associated with those cancers may have opposing effects on metabolism and still result in CC. Section 5.2 describes the literature concerning the effects of tumor derived exosomes on skeletal muscle while section 6.3 describes literature showing that tumor derived exosomes contribute to adipose tissue browning (which has much higher mitochondrial content and a greater metabolic rate). We have revised these sections to be more explicit about the effects of tumor derived exosomes on mitochondrial function. However, there is still much to learn about the impact of exosomes on mitochondria and vice versa.
Reviewer #1 comment. Is there any detail available regarding role of tumor or host derived exosomes on hypothalamus-pituitary axis?
Authors response: To our knowledge, there is no data that illustrates an effect of tumor derived exosomes on the hypothalamus-pituitary axis that results in a direct effect on skeletal muscle or adipose tissue. It is generally accepted that systemic inflammation as is typical of CC results in appetite suppression, which results in muscle and adipose atrophy because of unmet nutritional demand.
Host cell exosomes may regulate the hypothalamus-pituitary axis to some degree. Narasinhan et al. illustrate that human skeletal muscle in CC is enriched in miR-532-5p, which targets neuropeptide y 1 receptor (https://doi.org/10.1002/jcsm.12168). It is possible that this miR is secreted in exosomal cargo, however this study does not investigate this specifically and to our knowledge there are none published that do. The CC literature focuses primarily on the effects of tumor derived exosomes on adipocytes, consequently that is the focus of the adipocyte section of our review. However, in the context of type 2 diabetes Gao et al. show that adipocyte derived exosomes enter the hypothalamus and stimulate appetite through an mTOR signaling related mechanism (PMID: 31278836). We declined to include these studies in our review because they do not represent an action of tumor derived exosomes on these tissues.
Reviewer 2 Report
The review paper by Pitzer et al is a very interesting and well written manuscript. It covers an important field of research, i.e., cancer cachexia, from a novel and innovative, yet poorly unexplored, point of view. The concepts regarding pathogenesis of cancer cachexia and the role of inflammation as key driver is clear and well discussed in the paper. Also the construction and the organization of the text in the different paragraphs and section is adequate and clear. I have just few comments to be addressed before acceptance.
1) Page 2, line 52: This background is partially correct since muscle wasting occurs also as a direct consequences of cancer-related inflammation and inflammatory/catabolic mediators through several mechanisms. It cannot be view only as a compensatory mechanism, even if this theory is interesting. Otherwise the role of inflammation is very well recognized and described by the authors in the other parts of the manuscript.
2) page 3, paragraph 2.2.1.: please take into account that hypoxia is also a component of the immune inflammatory response in tumor microenvironment and strongly influence tumor and immune cells metabolism and then the pathogenesis of CC (see Cachexia as Evidence of the Mechanisms of Resistance and Tolerance during the Evolution of Cancer Disease. Int J Mol Sci. 2021 Mar 12;22(6):2890. doi: 10.3390/ijms22062890).
Author Response
Comments from Reviewer #2
Reviewer #2 comment. The review paper by Pitzer et al is a very interesting and well written manuscript. It covers an important field of research, i.e., cancer cachexia, from a novel and innovative, yet poorly unexplored, point of view. The concepts regarding pathogenesis of cancer cachexia and the role of inflammation as key driver is clear and well discussed in the paper. Also the construction and the organization of the text in the different paragraphs and section is adequate and clear. I have just few comments to be addressed before acceptance.
Authors response: Thank you for your encouraging comments in your review and helpful comments and suggestions for improving our manuscript.
Reviewer #2 comment. Page 2, line 52: This background is partially correct since muscle wasting occurs also as a direct consequences of cancer-related inflammation and inflammatory/catabolic mediators through several mechanisms. It cannot be view only as a compensatory mechanism, even if this theory is interesting. Otherwise the role of inflammation is very well recognized and described by the authors in the other parts of the manuscript.
Authors response: We have revised this statement to mention direct causes of muscle wasting mentioned above, while maintaining that muscle acts as an amino acid reservoir and may be wasted to meet energy demand.
Reviewer #2 comment. 2) page 3, paragraph 2.2.1.: please take into account that hypoxia is also a component of the immune inflammatory response in tumor microenvironment and strongly influence tumor and immune cells metabolism and then the pathogenesis of CC (see Cachexia as Evidence of the Mechanisms of Resistance and Tolerance during the Evolution of Cancer Disease. Int J Mol Sci. 2021 Mar 12;22(6):2890. doi: 10.3390/ijms22062890).
Authors response: Thank you for this suggestion. We have expanded on this paragraph to highlight the systemic effects of hypoxia, through HIF1 and the effects of tumor derived exosomes on the tumor microenvironment. Reconciling the systemic inflammation seen in CC with the local immunosuppressive nature of tumor derived exosomes in the tumor microenvironment.
Reviewer 3 Report
In this review, authors summarize the role of tumor-derived exosomes in cancer cachexia. They focused on the effects of exosome and exosomal components on muscle and adipose tissue metabolism and the associated biological processes. Overall, this review is well organized and prepared, and also completely summarizes the background and current studies regarding cancer-derived exosome and cancer cachexia with emphasis on skeletal muscle and adipose tissue wasting and inflammation. This review contributes to a clear understanding of the role of tumor-derived exosomes in cancer cachexia and its cellular acts. Only a minor issue, “miR” is more commonly used than “MiR” to represent microRNA.
Author Response
Comments from Reviewer #3
Authors comments. We appreciate the time and support of your review and suggestions for improving the manuscript.
Reviewer #3 comment. In this review, authors summarize the role of tumor-derived exosomes in cancer cachexia. They focused on the effects of exosome and exosomal components on muscle and adipose tissue metabolism and the associated biological processes. Overall, this review is well organized and prepared, and also completely summarizes the background and current studies regarding cancer-derived exosome and cancer cachexia with emphasis on skeletal muscle and adipose tissue wasting and inflammation. This review contributes to a clear understanding of the role of tumor-derived exosomes in cancer cachexia and its cellular acts. Only a minor issue, “miR” is more commonly used than “MiR” to represent microRNA.
Authors comments. We thank the reviewer for their constructive feedback. We have replaced every instance of “MiR” with “miR”.
Round 2
Reviewer 1 Report
Authors have answered the queries and modified the manuscript.